

# TarSynFlow, a workflow for bacterial genome comparisons that revealed genes putatively involved in the probiotic character of *Shewanella putrefaciens* strain Pdp11

Pedro Seoane[1], Silvana T. Tapia-Paniagua[2], Rocío Bautista[3],
Elena Alcaide[4], Consuelo Esteve[4], Eduardo Martínez-Manzanares[2],
M. Carmen Balebona[2], M. Gonzalo Claros[1,3] and Miguel A. Moriñigo[2]

[1] Department of Molecular Biology and Biochemistry, Universidad de Málaga, Málaga, Spain
[2] Department of Microbiology, Universidad de Málaga, Málaga, Spain
[3] Andalusian Platform for Bioinformatics, Universidad de Málaga, Málaga, Spain
[4] Department of Microbiology and Ecology, Universidad de Valencia, Valencia, Spain

## ABSTRACT

Probiotic microorganisms are of great interest in clinical, livestock and aquaculture. Knowledge of the genomic basis of probiotic characteristics can be a useful tool to understand why some strains can be pathogenic while others are probiotic in the same species. An automatized workflow called TarSynFlow (Targeted Synteny Workflow) has been then developed to compare finished or draft bacterial genomes based on a set of proteins. When used to analyze the finished genome of the probiotic strain Pdp11 of *Shewanella putrefaciens* and genome drafts from seven known non-probiotic strains of the same species obtained in this work, 15 genes were found exclusive of Pdp11. Their presence was confirmed by PCR using Pdp11-specific primers. Functional inspection of the 15 genes allowed us to hypothesize that Pdp11 underwent genome rearrangements spurred by plasmids and mobile elements. As a result, Pdp11 presents specific proteins for gut colonization, bile salt resistance and gut pathogen adhesion inhibition, which can explain some probiotic features of Pdp11.

## INTRODUCTION

Probiotics are living microorganisms, which, when administered in adequate amounts, confer a health benefit to the host (*Kechagia et al., 2013*). Very interesting results have been recently reported regarding fish benefits with respect to digestive enzymes, growth, and immune response when probiotics are included in feed (*Akhtar et al., 2015*; *Banerjee et al., 2017*). Probiotic candidates are screened and isolated from the indigenous microbiota of fish as an advantage over exogenous sources (*Boutin et al., 2013*). Different mechanisms have been demonstrated by probiotics and have been used to

Corresponding author
M. Gonzalo Claros, claros@uma.es

select them (*Ibrahem, 2013*), the antagonistic effect on pathogens being one of the most widely used (*Newaj-Fyzul, Al-Harbi & Austin, 2014*). However, the potential mechanisms involved in the probiotic character remains obscure.

*Shewanella putrefaciens* strain Pdp11 is a γ–Proteobacteria isolated from skin of farmed healthy gilthead seabream (*Sparus aurata*) (*Chabrillón et al., 2006*) that has shown beneficious effects on farmed gilthead seabream and Senegalese sole (*Solea senegalensis*). Benefits include growth improvement (*Sáenz de Rodrigáñez et al., 2009*), resilience against stress (*Varela et al., 2010*), immunological response (*Díaz-Rosales et al., 2009*; *Tapia-Paniagua et al., 2015*) and resistance against diseases (*Díaz-Rosales et al., 2009*; *Tapia-Paniagua et al., 2014*). As a logical consequence, this microorganism has been proposed as a probiotic for the farming of Senegalese sole and gilthead seabream (*Tapia-Paniagua et al., 2012*). Similar benefits are provided by probiotic *Shewanella colwelliana* WA64 and *Shewanella olleyana* WA65 by enhancing innate immunity, respiratory activity, protein levels and disease resistance of abalone, a marine shellfish, in intensive culture (*Jiang et al., 2013*). Interestingly, *Shewanella putrefaciens* also includes pathogenic and saprophytic strains with relevance to fish spoilage and fish infection (*Esteve, Merchán & Alcaide, 2016*). The recent sequencing of its genome (*Tapia-Paniagua et al., 2017*) provides an opportunity to discern the genetic bases of its probiotic character based on a comparison between genomes of probiotic, pathogenic and saprophytic strains.

Recent advances in next-generation sequencing technologies and the small size of the bacterial genomes have promoted the development of a huge amount of sequencing projects in this area (*Tatusova et al., 2014*). In fact, sequencing several strains from the same species can explain phenotype differences based on genetic changes (*Boucher, Nesbø & Doolittle, 2001*). Hence, many tools have been developed to compare genome sequences and to understand new sequenced genomes, such as SynChro (*Drillon, Carbone & Fischer, 2014*), DRIMM-Synteny (*Pham & Pevzner, 2010*), and Sibelia (*Minkin et al., 2013*). In some cases, graphical (Mauve (*Darling et al., 2004*), MizBee (*Meyer, Munzner & Pfister, 2009*), and SyMap (*Soderlund, Bomhoff & Nelson, 2011*)) or web (Synteny Portal (*Lee et al., 2016*), SyntTax (*Oberto, 2013*), and SynTView (*Lechat et al., 2013*)) user-friendly interfaces are available. Moreover, Sibelia, SyntTax and SynTView are devoted to prokaryotic genomes and use similarity algorithms to perform a synteny analyses revealing changes or rearrangements from one genome to another (*Lemoine, Lespinet & Labedan, 2007*). Highly similar shared regions are usually regarded as conserved blocks revealing the synteny. This criterion is very useful for evolution analysis, when the interest is focused on the relation between genomes, but it is not suitable when small differences between highly syntenic genomes is the focus.

Synteny studies established that bacterial genomes are highly dynamic (*Rocha, 2004*) and close species, even strains (*Boucher, Nesbø & Doolittle, 2001*), have enormous sequence rearrangements. Since these analyses only provide conservation of genome blocks, they usually disregard the functional information inferred from differences between the compared genomes. Moreover, they are usually limited to tracking a few genes related to a particular biological problem. Hence, a comparison of the completely sequenced probiotic

PeerJ ────────────────────────────────────────

strain *Shewanella putrefaciens* Pdp11 against several non-probiotic strains (NPSs) of this species would be expected to provide functional information about the genetic basis of some of its probiotic features. Consequently, genome drafts of five pathogenic and two saprophytic strain of *Shewanella putrefaciens* were obtained and contrasted to the probiotic strain Pdp11. A workflow called TarSynFlow (Targeted Synteny workFlow) was developed to perform a targeted but comprehensive similarity searches between bacterial genomes. The comparison provided genome location and functional annotation of 15 Pdp11-specific proteins, likely related to the colonization capabilities of *Shewanella putrefaciens*. The experimental validation by PCR confirmed the suitability of the TarSynFlow design.

## MATERIALS AND METHODS

### Bacteria and growth conditions

In this study eight separate isolates binned to be *Shewanella putrefaciens* by their 16S barcoding (*Esteve, Merchán & Alcaide, 2016*) were used to evaluate in silico the putative genes that might be involved in some probiotic features. One isolate, Pdp11, was established as probiotic for farmed fish (*Tapia-Paniagua et al., 2012*); two saprophytic isolates (SdM1 and SdM2) were identified in environmental sources; and five isolates (SH4, SH6, SH9, SH12 and SH16) were pathogenic for eel (*Esteve, Merchán & Alcaide, 2016*). All isolates were grown in trypticase soy agar (TSA; Merck, Darmstadt, Germany) with 1.5% sodium chloride (w/v), for 24 h at 23 °C, aerobically, as a pure culture. Pathogenic isolates were characterized (Table 1) based on their values of LD50 (dose which is lethal to 50% of bacterial population as determined in specimens of *Anguilla anguilla*), as well as random amplification of polymorphic DNA (RAPD) profiles and the growth at 6% NaCl and 37 °C, as described by *Esteve, Merchán & Alcaide (2016)*.

### DNA isolation, sequencing and assembly of draft genomes

One colony of every NPS was grown to exponential phase in TSBs (Tryptone Soy Broth, Oxoid) supplemented with 1.5% sodium chloride and then centrifuged (2,500*g*, 15 min). Pellets were washed with phosphate-buffered saline (PBS) and used for DNA extraction according to the manufacturer's instructions (Thermo Scientific, Schwerte, Germany). DNA was suspended in 100 µl of molecular biology water and stored at 4 °C. DNA quality and yield were analyzed by agarose (1%, w/v) gel electrophoresis loading the samples with RedSafe™ Nucleic Acid Staining Solution (Sigma-Aldrich, St. Louis, MO, USA). Fluorometric quantification of DNA was performed by Qubit system (Thermo Scientific, Germany).

DNA from all these were sequenced in a single run using the Illumina MiSeq platform at the sequencing service of Centro de Investigaciones Médico-Sanitarias (CIMES) (University of Malaga). The sequencing library was built with the Nextera protocol and the Illumina kit 2 × 300 bp, and raw reads are available at BioProject PRJNA510237. Raw reads were pre-processed and assembled using the A5-miseq pipeline (*Coil, Jospin & Darling, 2015*) with default parameters. Assembling completeness was determined using

**Table 1 Microbiological characterization of S. putrefaciens pathogenic strains used in this study.**

| Pathogenic strain | LD50 (cfu/g) | Profile of RAPD | Growth at 6% NaCl | Growth at 37 °C |
|---|---|---|---|---|
| SH4 | $3.4 \times 10^6$ | I | + | + |
| SH6 | $8.3 \times 10^6$ | ND | − | + |
| SH9 | $1.4 \times 10^6$ | II | + | + |
| SH12 | $2.8 \times 10^6$ | III | − | + |
| SH16 | $5.5 \times 10^6$ | III | + | − |

Benchmarking Universal Single-Copy Orthologs (BUSCO) (*Simão et al., 2015*) with the 452 proteins of the γ-Proteobacteria database provided with the software.

### *Shewanella putrefaciens* sequences

The full genome of the probiotic strain Pdp11 of *Shewanella putrefaciens* was obtained from the NCBI project ID PRJNA312231 (accession number CP015194.1). A total of 8,171 protein sequences of *Shewanella putrefaciens* were retrieved from UniProtKB to date July of 2016 using "*Shewanella putrefaciens*" as organism keyword. Sequence redundancy was removed during TarSynFlow execution to yield 46.19% of the retrieved sequences as unique, resulting in a *Shewanella putrefaciens* protein reference set containing 3,774 sequences of amino acids (File S1).

### TarSynFlow workflow description

TarSynFlow (Targeted Synteny workFlow) is customizable workflow based on our workflow manager AutoFlow (*Seoane et al., 2016*) for Linux/UNIX based supercomputers, that can be downloaded from https://github.com/seoanezonjic/TarSynFlow. Its execution requires the installation of other tools (see the *Readme.md* file), such as CD-HIT (*Li & Godzik, 2006*) for sequence redundancy removal; BLAST (*Camacho et al., 2009*) for similarity searches; PROSPLIGN (*Kapustin et al., 2008*) for polishing gene boundaries; and CIRCOS (*Krzywinski et al., 2009*) for graphic representation. Additionally, Ruby gems *scbi_distributed_blast* (*Guerrero-Fernández, Falgueras & Claros, 2013*) and *make_circos* (this work) must also be installed.

Three sequence files in FastA format are required to launch TarSynFlow (Fig. 1): two bacterial genomes, A and B, and the protein reference file containing the amino acid sequences of the set of proteins to be compared and located in both A and B genomes. Threshold customization for protein-identity and protein coverage that will split similarity profiles among "high-similarity" and "low-similarity" is allowed. By default, the high-similarity profile would contain protein matches having >85% identity and >85% coverage thresholds, while both thresholds are decreased to 45% for the low-similarity profile. Both profiles are useful for the evaluation of the significance of the protein matches obtained along the analysis. More details about workflow configuration is given in the above mentioned *Readme.md* file.

TarSynFlow starts removing sequence redundancy in the protein reference using CD-HIT, with an identity threshold of 60%, to produce a non-redundant reference (Fig. 1)

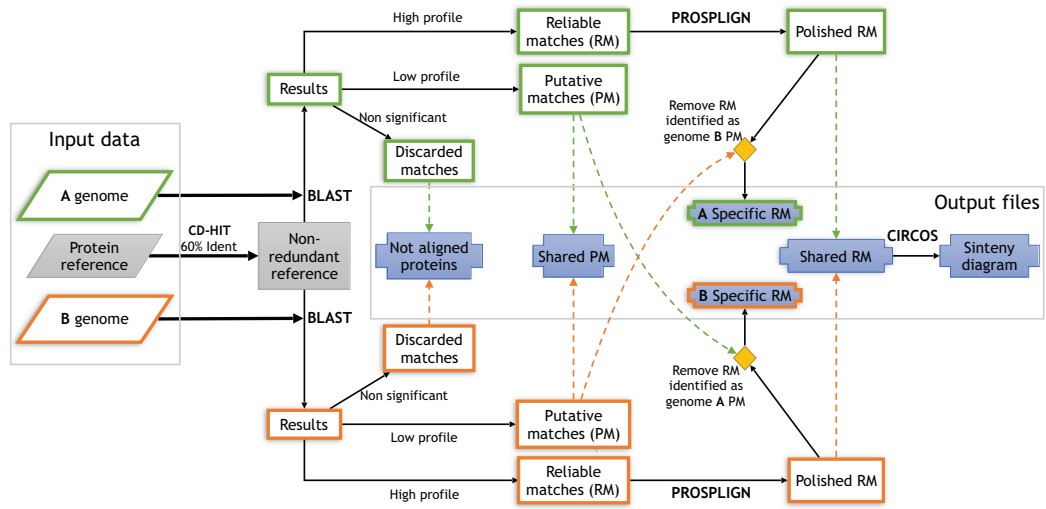

**Figure 1 Flow chart describing the TarSynFlow workflow.** Solid lines represent analyses performed for every protein in a specific genome, where green box-lines depict processes applied to proteins of genome A and orange box-lines depict those for the other genome. Boxes outputting results in files are in solid blue, encircled with green or orange box-lines when are genome-specific, and with a thin blue box-line for comparative results. Dashed lines represent the comparison of protein IDs for the two genomes, with the line colour indicating genome A or B source. 'CD-HIT', 'Blast', 'Prosplign' and 'Circos' are in bold uppercase because they correspond to third party software. 'High profile' refers to the filter that keeps only protein matches with protein coverage and identity ≥85%, while 'Low profile' refers to the one keeping also protein matches with protein coverage and identity between 85% and 45%. See text for further details.

that minimizes overlapping matches. The non-redundant protein reference is sent to simultaneous BLASTX against the A and B genomes. The resulting hits are then filtered out using the high-similarity profile to collect the set of Reliable Matches (RM) to be processed with PROSPLIGN to polish gene boundaries and fine-tuning the exact gene coordinates. A table with the common RM in both genomes and their coordinates is then provided to be represented with CIRCOS as a typical circular diagram (as in Fig. 2). The remaining BLASTX hits are filtered again using the low-similarity profile (Fig. 1) to obtain the Putative Matches (PM). Accordingly, common proteins for genomes A and B are those having PM or RM qualification with the two genomes, while genome-specific proteins are those having a RM with this genome, but no RM nor PM with the other genome. Finally, five protein ID sets are saved: (1) the most reliable set of proteins shared by both genomes, corresponding to high profile protein matches, (2) the less reliable set of shared proteins, where each protein provides a PM with at least one genome, (3) the highly reliable genome A-specific proteins, (4) the highly reliable genome B-specific proteins and (5) not aligned, or not reliably aligned, reference proteins to any of the analyzed genomes.

## Extending the comparison to more than two genomes

The capabilities of TarSynFlow can be extended to a multiple comparison of one test genome against a set of different genomes. To do so, the complete set of paired comparisons of the test genome against the other genomes is mined using the bash script

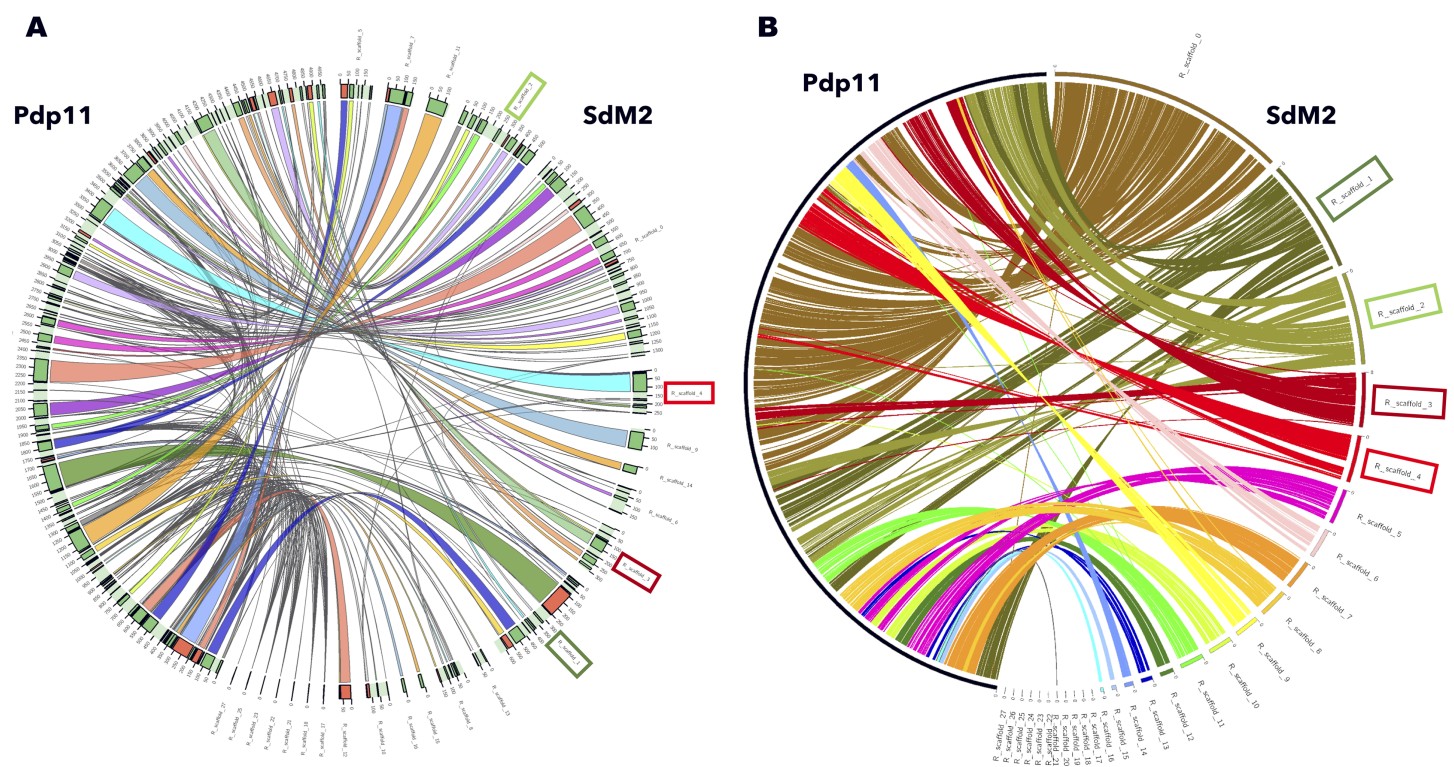

**Figure 2 Synteny diagrams of probiotic Pdp11 and SdM2 saprophytic strain as CIRCOS output.** Data for synteny were obtained using Sibelia (A) with a synteny block length of 500 nt in order to generate comparable results with TarSynFlow (B). Exemplary scaffolds with sequence rearrangements are boxed with the same color in A and B panels for comparison.

*get_all_results.sh* (present in the GitHub repository of TarSynFlow) to summarize the shared and different reference proteins of the test genome with respect to the others. In detail, *get_all_results.sh* compares the saved protein ID lists of every TarSynFlow execution and generates a new list containing the protein IDs identified across all paired comparisons. The minimum number of occurrences to consider a protein match as significant can be customized (by default, all the paired comparisons, seven in this work). Functional information of IDs is then retrieved using UniProt web services to obtain description, gene name, amino acid length, review status, source organism and GO terms. To reveal possible presence/absence correlation patterns between strains and specific proteins, those proteins are then clustered as a heatmap using the *gplots* package for R.

## DNA extraction and PCR

Total DNA was extracted from each strain as described above. Primers for genes considered as unique in Pdp11 (Table 2) were designed using Primer3 software (freely available at http://bioinfo.ut.ee/primer3-0.4.0/) according to locations in the Pdp11 genome.

PCR amplification was performed using 20 ng of genomic DNA from the different strains or isolates in a total volume of 20 μl containing 12 μl of SsoAdvanced™ Universal SYBR® Green Supermix (Bio-Rad Laboratories, Hercules, CA, USA) and 10 mM

**Table 2 PCR primers designed to verify the probiotic-specific genes (described by their UniProt ID) based on the sequence of Pdp11.**

| Uniprot ID | Primer | Sequence | Mt (°C) | Size (pb) | AnnT (°C) |
|---|---|---|---|---|---|
| E6XIZ3 | Z3-Pdp11-F | TCAGGGTCTTCGAATCTTCC | 59.9 | 1,344 | 53 |
| | Z3-Pdp11-R | AGAGCAGCACAGTCAAAGCA | 59.2 | | |
| E6XIZ2 | Z2-Pdp11-F | TTGCTGTTGTTGGTGGGTTA | 60.0 | 2,235 | 55 |
| | Z2-Pdp11-R | AGCGTTTAGCCGAACTTGAA | 60.0 | | |
| E6XG14 | G14-Pdp11-F | AACCGAGCAGTGCATTTTCT | 58.4 | 1,294 | 53 |
| | G14-Pdp11-R | CACACCGTCAGTTCCAAAAT | 59.8 | | |
| E6XG15 | G15-Pdp11-F | TGCATACCGCGAACTAAGTG | 58.9 | 1,252 | 55 |
| | G15-Pdp11-R | CAGATAAGCCATGAAGCAACA | 59.9 | | |
| E6XIZ5 | Z5-Pdp11-F | CCTGAAAACGCACCAAGTTT | 59.9 | 1,007 | 53 |
| | Z5-Pdp11-R | CAGCAGTAAAATGACGCAACA | 60.1 | | |
| O86914 | 6914-Pdp11-F | CAAACCCAATACGGTCCATC | 60.0 | 2,058 | 55 |
| | 6914-Pdp11-R | GCTGACCTTAGGCACTTTGC | 60.0 | | |
| E6XL69 | L69-Pdp11-F | CATCCAAAGGATTTAATTTAAGTGG | 60.1 | 575 | 53 |
| | L69-Pdp11-R | GTGATACCTAGGGCGACGAA | 59.2 | | |
| E6XIZ4 | Z4-Pdp11-F | GGTTACATCATATTCTCTGCATGAT | 59.0 | 585 | 53 |
| | Z4-Pdp11-R | GTAACTCCCCAATTGCAGAAA | 58.5 | | |
| E6XLE5 | LE5-Pdp11-F | GGCTTAACAATCACGCCAAT | 58.0 | 473 | 53 |
| | LE5-Pdp11-R | ATGTCCGGATGCTACAAAAA | 59.9 | | |
| Q8GJK1 | K1-Pdp11-F | TCGGTTACCATTTACTCTCAGC | 58.4 | 905 | 55 |
| | K1-Pdp11-R | GGAGATGTTTTTGTGTCGTGTT | 59.1 | | |
| Q6ZYR2 | R2- Pdp11-F | TGAGCCAACCCAATCTATCC | 59.8 | 1,085 | 55 |
| | R2-Pdp11-R | GTGGCAACCTCTTCTTGTCC | 60.0 | | |
| A4Y1U2 | U2-Pdp11-F | ACACCAGTTGGGCGATAAAA | 60.0 | 873 | 54 |
| | U2-Pdp11-R | ATCGGCAAGGTTTAAAAGCA | 59.7 | | |
| A4Y11U5 | U5-Pdp11-F | CCAGTCACCACACTCATTGG | 60.0 | 1,932 | 55 |
| | U5-Pdp11-R | GCTTATGAACGCACCCGTAT | 59.9 | | |
| A1KQX7 | X7-Pdp11-F | TACCTGGATGAAATGCGTCA | 55.1 | 500 | 57 |
| | X7-Pdp11-R | TCGTGTTTCGATAAGGCTGA | 55.1 | | |
| A4Y11U4 | U4-Pdp11-F | TCGACGATCATCATCTGAGAA | 59.8 | 575 | 54 |
| | U4-Pdp11-R | TTCAGCTGATGCATACCAAAG | 58.9 | | |
| A4YB89 | B89-Pdp11-F | GCCATCATAGGCGAGCTAAC | 60.2 | 900 | 54 |
| | B89-Pdp11-R | ATCAACTGCATGACAATAAAAACG | 59.8 | | |

**Note:**
The melting temperature (Mt) for every primer, as well as the amplicon size and the annealing temperature (AnnT) for every primer pair are given. F, Forward primer; R, Reverse primer.

each primer described in Table 2). Amplification was made in triplicate for each sample and carried out in a CFX96 Touch™ Real-Time PCR Detection System (BioRad, Hercules, CA, USA). The following conditions were applied: 95 °C for 3 min, followed by 28 cycles of 95 °C for 30 s, annealing temperature depending on the primer pair (Table 2) for 30 s, and 72 °C for 30 s for short sequences and 1.30 min for the largest, and a final cycle of 72 °C for 5 min. PCR products were analyzed by 1% (w/v) agarose gel electrophoresis stained with RedSafe™ to check the products for the expected size.

**Table 3** A5-miseq summary for sequencing and assembling data for the NPSs used in this study.

| Strain | Raw reads | Useful reads (%) | N50 (bp) | Genome size (bp) | %GC | Scaffold number | Completeness (%) |
|--------|-----------|------------------|----------|------------------|------|-----------------|-------------------|
| SH4    | 2,303,512 | 98.50 | 223,087 | 4,628,646 | 46.3 | 46 | 99.4 |
| SH6    | 2,047,622 | 98.28 | 259,802 | 5,022,912 | 45.3 | 44 | 99.5 |
| SH9    | 1,193,322 | 96.16 | 245,702 | 5,020,097 | 45.3 | 47 | 99.5 |
| SH12   | 1,650,318 | 98.03 | 160,200 | 4,628,973 | 46.3 | 58 | 99.4 |
| SH16   | 2,319,174 | 98.21 | 387,271 | 5,018,364 | 45.3 | 37 | 99.5 |
| SdM1   | 3,262,744 | 98.33 | 347,522 | 5,068,163 | 45.2 | 44 | 99.3 |
| SdM2   | 4,227,076 | 98.57 | 511,212 | 4,354,804 | 44.3 | 28 | 99.1 |

## Sequencing of amplified PCR products

DNA from amplicons was purified using GeneJET PCR Purification Kit (BioRad, Hercules, CA, USA) according to the manufacturer's instructions and then sequenced at Macrogen (Seul, South Korea). The quality control of the sequences was performed using the percentage of bases with a quality score higher than 20 (reported by Macrogen). Then, the sequences were compared with the complete genome of Pdp11 using BLASTN (*Camacho et al., 2009*).

## RESULTS

### Genome drafts for non-probiotic strains of *Shewanella putrefaciens*

With the aim of detecting genes or genomic regions from the probiotic strain that are distinctive with respect to NPSs, the recently published (*Tapia-Paniagua et al., 2017*) genome draft of *Shewanella putrefaciens* Pdp11 in one single scaffold was used. The circular genome consists of 4.973 Mb (GenBank AC# CP015194.1), which is similar in size to other finished *Shewanella sp.* genomes that range from 4.706 Mb for MR-4 (assembly GCA_000014685.1) to 5.266 MB for WE21 (assembly GCA_002966515.1) as summarized in the NCBI page https://www.ncbi.nlm.nih.gov/genome/genomes/13542.

As a source of different instances of *Shewanella putrefaciens* NPSs, seven bacterial isolates previously classified as *Shewanella putrefaciens* by 16S barcoding (*Esteve, Merchán & Alcaide, 2016*) were used. According to the standard tests shown in Table 1, five of them (SH4, SH6, SH9, SH12 and SH16) were then considered different pathogenic strains, while the other two (SdM1 and SdM2) were considered different saprophytic isolates. Their genomes were sequenced, the reads were pre-processed and then assembled to provide a number of scaffolds that ranged from 28 for SdM2 to 58 for SH12 (Table 3). These figures are quite acceptable since only 11 ongoing sequencing projects for *Shewanella sp.* are in 26–57 scaffold range, while up to 26 are in the 61–1,135 scaffold range. As expected, the N50 increases as the number of scaffold decreases, although for the same number of scaffolds (44), SdM1 is more contiguous than SH6. Moreover, NPS genome size, ranging from 4.35 to 5.068 Mb, and GC content, ranging from 44.3% to 46.3%, were also in agreement with genome sizes and %GC of Pdp11 and other finished *Shewanella sp.* as appear in https://www.ncbi.nlm.nih.gov/genome/genomes/13542. Completeness of the seven NPS

**Table 4 Summary of protein matches revealed by TarSynFlow when Pdp11 was the test genome compared to the NPSs of Table 3.**

| NPS name | NPS-specific | Pdp11-specific | Shared by Pdp11 & NPSs | Probably shared by Pdp11 & NPSs | Not assigned to any strain |
|---|---|---|---|---|---|
| SH4 | 79 | 90 | 1,930 | 1,219 | 1,330 |
| SH6 | 130 | 41 | 2,286 | 959 | 1,234 |
| SH9 | 130 | 41 | 2,285 | 960 | 1,234 |
| SH12 | 79 | 91 | 1,931 | 1,217 | 1,331 |
| SH16 | 130 | 41 | 2,286 | 959 | 1,234 |
| SdM2 | 305 | 29 | 2,333 | 946 | 1,200 |
| SdM1 | 116 | 43 | 2,277 | 964 | 1,238 |
| Common to all | 64 | 19 | 1,886 | 834 | 1,160 |

genome drafts was estimated using BUSCO and was quite high, ranging from 99.1% to 99.5% (Table 3, last column). Since gaps among scaffolds are likely due to the presence of repeated sequences—although non-covered genome sequences cannot be discarded—these genome drafts, although slightly fragmented, seem suitable for further comparative analyses using TarSynFlow, especially because probiotic, pathogenic and saprophytic characters are expected to lie within in non-repetitive sequences (the part evaluated by BUSCO).

## TarSynFlow provides a dense and reliable set of links between genomes

TarSynFlow locates every protein sequence from the non-redundant protein reference in the two compared genomes and classifies proteins between common to (shared by) both genomes and specific only for one of the genomes. To show the potential of this comparative approach, TarSynFlow was compared to Sibelia (*Minkin et al., 2013*), a widely used tool in bacterial synteny based on DNA sequence comparisons. The test genome was Pdp11 (*Tapia-Paniagua et al., 2017*) and the other genome was SdM2 since it presents the lower number of scaffolds in Table 3, being therefore the less fragmented genome draft. Both Sibelia and TarSynFlow results were then plotted using CIRCOS (Fig. 2). Ribbons in Fig. 2A produced by Sibelia show 1,384 synteny blocks and many gaps without connections, where gaps are segments where the nucleotide sequence from both genomes, even if they are syntenic, are more divergent. Nevertheless, TarSynFlow produces a denser relation (2,355 high-similarity profile links, discarding the low-similarity profile links to produce consistent matches) between both genomes (Fig. 2B) since they are based on protein similarity instead of nucleotide identity. Since each TarSynFlow link is based on a gene-coded protein and not simply in nucleotide sequence similarity or identity, the ribbons are wider, producing a more realistic synteny overview. Also, putative rearrangements can be easily inferred from Fig. 2B (for instance, scaffolds 1, 2, 3 and 4 of SdM2, highlighted with boxes), disregarding whether these rearrangements were derived from misassembling or real genome rearrangements.

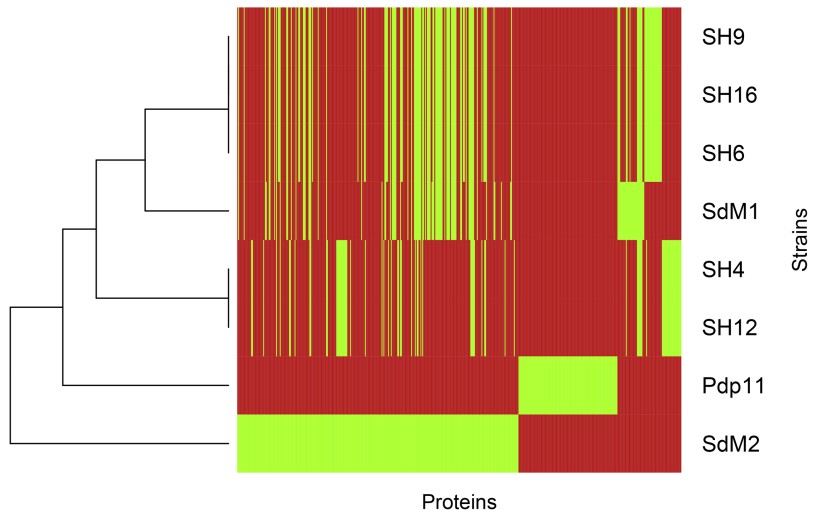

**Figure 3 Strain clustering based on differential proteins.** Proteins which are differentially present or absent in genomes were clustered by their pattern of presence and absence in the eight strains analyzed in this study. Green: the protein is present; red: the protein is absent.

## Gene differences between Pdp11 and non-probiotic strains

Table 4 summarizes the results after TarSynFlow comparison of Pdp11 against each one of the seven NPS genomes in Table 3. The number of specific proteins for each NPS ranged from 79 for SH4 and SH12 to 305 in SdM2, with 64 NPS-specific proteins. Pdp11-specific proteins ranged from as low as 29 with SdM2 to 91 with SH12, where 19 appeared in all cases an can be considered the feeding set of Pdp11-specific proteins related to its probiotic character. It is also shown that Pdp11 and NPSs share from 1,930 to 2,333 proteins, with 1,886 proteins being common for all strains (Pdp11 and NPSs). This number may be increased if the 834 proteins providing PM are considered (Table 4), rendering a total of 2,720 proteins shared by all strains analyzed. This confirms that, although strain-specific genes are present, most genes are conserved between strains, in agreement with the many links connecting Pdp11 with SdM2 in Fig. 2.

## Apparent strain groups within non-probiotic strains

The figures in Table 4 drove us to split pathogenic strains into two groups with very homogeneous and separate number of shared/specific proteins: one group consisting of SH4 and SH12 presents (i) 79 NPS-specific proteins, (ii) 90–91 Pdp11-specific proteins, (iii) 1,930–1,931 certainly-shared proteins, (iv) 1,217–1,219 probably-shared proteins and (v) 1,330–1,331 unaligned proteins. The other group is comprised of SH6, SH9 and SH16, and presents (i) more NPS-specific proteins (130), (ii) less Pdp11-specific proteins (41), (iii) more certainly-shared proteins (2,285–2,286), (iv) less probably-shared proteins (959–960) and (v) less unaligned proteins (1,234).

Regarding the saprophytic strains SdM1 and SdM2, they present numbers quite different to pathogenic strains, even though SdM2 displays the highest number of NPS-specific proteins (305) and the lower number of Pdp11-specific proteins (29). When

shared protein IDs are used to classify the different strains (Fig. 3), it is clearly seen that the proposed groups are consistent, with SdM1 closer to pathogenic strains, and SdM2 closer to Pdp11. This indicates that saprophytic strains are more heterogeneous than pathogenic strains. In fact, the saprophytic strains present the lower rate of shared protein IDs between both strains (about 75% for the specific saprophytic proteins and 72.41% for the specific Pdp11 proteins). Similar sharing ratios were found when shared and not-aligned protein categories, as can be deduced from Fig. 3. In fact, Fig. 3 also supports the grouping of SH4 and SH12 since they are nearly identical from the protein-sharing point of view, in spite of microbiological data in Table 1 that indicate that they are different isolates.

It is worth noting that strain group SH4-SH12 presents a high number of scaffolds (58 for SH12 and 46 for SH4, Table 3), less NPS-specific proteins and the highest Pdp11-specific proteins (Table 4). This can be explained by genome divergence or by gene information lost due to assembling gaps, even though their completeness is 99.4%. In fact, it can be hypothesized that 1,160 proteins of the non-redundant reference that are not present in any of the genomes analyzed are absent in Pdp11 and may be present in gaps within genome drafts.

## Pdp11-specific sequences are experimentally absent in non-probiotic strains

The seven paired comparisons of Pdp11 with the genome drafts of Table 3 were compared with the script *get_all_results.sh*, setting seven as the minimum occurrence number to consider a protein as significant. The results (Table 4) provide 19 Pdp11-specific proteins whose details are presented in Table 5, and 64 NPS-specific proteins whose details appear in File S2. From Pdp11-specific proteins, A4L329, Q70IK8 and Q70IK5 (tagged with an asterisk at the end of Table 4) were discarded for experimental validation due to the presence of the tag "Fragment" in the annotating orthologue in UnitProtKB. Since among the remaining 16 Pdp11-specific proteins (the first 16 rows in Table 4) should reside genes contributing to probiotic features of Pdp11, their presence in Pdp11 and their absence in the other strains was experimentally validated by PCR. Figure 4 illustrates the PCR amplification of three candidates, while the results for the complete set of genes are summarized in Table 6. Only the gene sequence for O86914 (trimethylamine *N*-oxide reductase) shows unspecific amplification in the strain group SH4-SH12, and was discarded. Interestingly, this finding is in agreement with the NPS grouping described above. Since all amplicons showed the expected size, only six of them (E6XG15, E6XG14, Q8GJK1, Q6ZYR2, A4Y1U2 and E6XLE5) were randomly selected for sequencing. The resulting sequence was compared to the Pdp11 genome by means of BLASTN, obtaining an identity minimum of 78.41 % and maximum of 99.71% with Pdp11 (Table 6), where identity divergences were caused by low quality of sequencing rather than true nucleotide changes (File S3). Therefore, the in silico prediction of 15 Pdp11-specific proteins (not present in NPSs) was experimentally confirmed, demonstrating that their absence in the NPSs is not an artefact due to the draft nature of the NPS genomes.

**Table 5 Specific UniProt IDs for the probiotic strain Pdp11 and absent in the NPSs.**

| UniProt ID | Protein length | Protein description | Gene ontology terms |
|---|---|---|---|
| E6XIZ3 | 360 | Bile acid/detergent exporter membrane fusion component, VexC | Membrane [GO:0016020]; transmembrane transport [GO:0055085] |
| E6XIZ2 | 1,011 | Bile acid/detergent exporter permease component, VexD | Integral component of membrane [GO:0016021]; transporter activity [GO:0005215] |
| E6XG14 | 353 | Undecaprenyl-phosphate alpha-N-acetylglucosaminyl 1-phosphate transferase | Gram-negative-bacterium-type cell wall [GO:0009276]; integral component of plasma membrane [GO:0005887]; magnesium ion binding [GO:0000287]; manganese ion binding [GO:0030145]; phospho-N-acetylmuramoyl-pentapeptide-transferase activity [GO:0008963]; transferase activity, transferring glycosyl groups [GO:0016757]; UDP-N-acetylglucosamine-undecaprenylphosphate N-acetylglucosaminephosphotransferase activity [GO:0036380]; O antigen biosynthetic process [GO:0009243] |
| E6XG15 | 357 | Undecaprenyl-phosphate alpha-N-acetylglucosaminyl 1-phosphate transferase | Gram-negative-bacterium-type cell wall [GO:0009276]; integral component of plasma membrane [GO:0005887]; magnesium ion binding [GO:0000287]; manganese ion binding [GO:0030145]; phospho-N-acetylmuramoyl-pentapeptide-transferase activity [GO:0008963]; transferase activity, transferring glycosyl groups [GO:0016757]; UDP-N-acetylglucosamine-undecaprenylphosphate N-acetylglucosaminephosphotransferase activity [GO:0036380]; O antigen biosynthetic process [GO:0009243] |
| E6XIZ5 | 242 | MltA-interacting MipA family protein | |
| O86914 | 829 | Trimethylamine-N-oxide reductase | Periplasmic space [GO:0042597]; electron carrier activity [GO:0009055]; molybdenum ion binding [GO:0030151]; trimethylamine-N-oxide reductase (cytochrome c) activity [GO:0050626]; trimethylamine-N-oxide reductase activity [GO:0009033] (EC 1.6.6.9) |
| E6XL69 | 105 | Putative uncharacterized protein | Integral component of membrane [GO:0016021] |
| E6XIZ4 | 111 | Putative uncharacterized protein | |
| E6XLE5 | 60 | Putative uncharacterized protein | |
| Q8GJK1 | 215 | HTH-type transcriptional regulator for conjugative element pMERPH | Sequence-specific DNA binding [GO:0043565]; regulation of transcription, DNA-templated [GO:0006355]; transcription, DNA-templated [GO:0006351] |
| Q6ZYR2 | 413 | Putative integrase | DNA binding [GO:0003677]; DNA integration [GO:0015074]; DNA recombination [GO:0006310] |
| A4Y1U2 | 206 | Resolvase, N-terminal domain | DNA binding [GO:0003677]; recombinase activity [GO:0000150] |
| A4Y1U5 | 1,026 | Transposase Tn3 family protein | transposase activity [GO:0004803]; transposition, DNA-mediated [GO:0006313] |
| A1KQX7 | 66 | Putative excisionase (Recombination directionality factor) | |
| A4Y1U4 | 103 | Plasmid stabilization system | |
| A4YB89 | 222 | Transposase | |
| A4L329* | 48 | TraG (Fragment) | |

| UniProt ID | Protein length | Protein description | Gene ontology terms |
|---|---|---|---|
| Table 5 (continued). | | | |
| Q70IK8* | 194 | Putative transfer protein (Fragment) | |
| Q70IK5* | 70 | Putative conjugative transfer protein (Fragment) | |

**Notes:**
Data are as output from the *get_all_results.sh* script for the comparative analysis of the seven TarSynFlow executions (one per NPS, using Pdp11 as the test genome).
\* Orthologue containing the tag "Fragment" within metadata.

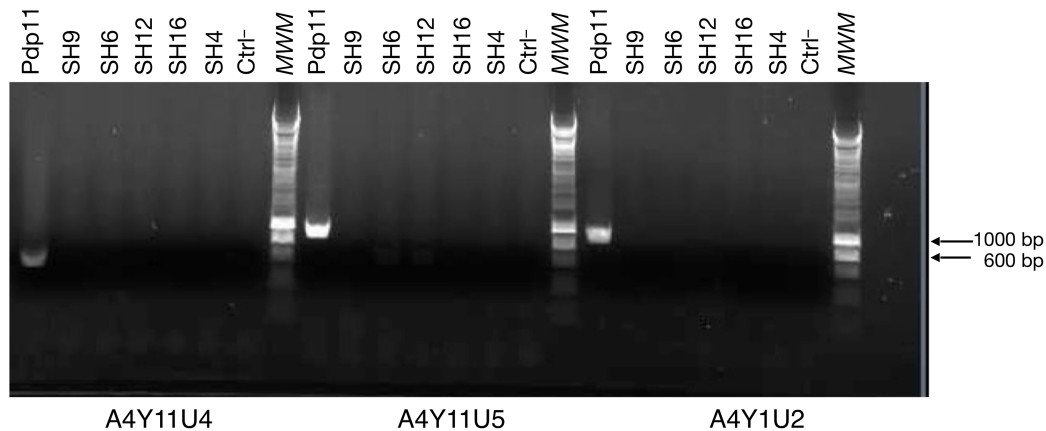

**Figure 4 Example of PCR amplification for three genomic sequences predicted to code Pdp11-specific proteins.** *MWM* is the molecular weight marker; arrows indicate bands for 600 and 1,000 bp. Ctrl⁻ indicates a negative control without DNA. In the three cases, amplification was obtained only in Pdp11, which confirms the in silico prediction that these genes are absent in the NPSs and are not an artefact due to the draft nature of the genomes of the NPSs.

## DISCUSSION

### TarSynFlow facilitates bacterial genome comparisons

The bioinformatic workflow TarSynFlow (Fig. 1) was designed to enable whole genome comparison between related species with the aim of locating conserved and distinctive gene-containing genome regions (Fig. 2). TarSynFlow capabilities have been illustrated with the in silico detection (Table 5) and experimental confirmation (Table 6) of genes that are present in the probiotic strain *Shewanella putrefaciens* Pdp11 with respect to seven NPSs of the same species (Table 1) that have been sequenced and assembled in this work (Table 3). The use of the bash script *get_all_results.sh* allows to overcome the limitation of paired-genome comparisons inherent to the TarSynFlow code, and several genomes can be compared against the same "test genome" (Pdp11 in this work). This enabled a reliable determination of present/absent protein orthologues in Pdp11 and all NPS genome drafts analyzed.

Although finished genomes are a priori preferred for comparative analyses, results of Tables 4 and 5 and File S2, as well as the experimental validation shown in Fig. 4, demonstrated that TarSynFlow can deal with draft versions of at least one, or even both, compared genomes. The drawback of draft usage lies in the possible overlooking of existing

Table 6 PCR validation of sequences coding for Pdp11-specific proteins using the primer pairs of Table 2 for Pdp11-sequences coding Pdp11-specific proteins in Table 5.

| Protein ID | Isolated strains | | | | | | | |
|---|---|---|---|---|---|---|---|---|
| | SH4 | SH6 | SH9 | SH12 | SH16 | SdM1 | SdM2 | Pdp11 |
| E6XIZ3 | – | – | – | – | – | – | – | + |
| E6XIZ2 | – | – | – | – | – | – | – | + |
| E6XG14 | – | – | – | – | – | – | – | + (96.73%) |
| E6XG15 | – | – | – | – | – | – | – | + (98.25%) |
| E6XIZ5 | – | – | – | – | – | – | – | + |
| O86914 | + | – | – | + | – | – | – | + |
| E6XL69 | – | – | – | – | – | – | – | + |
| E6XIZ4 | – | – | – | – | – | – | – | + |
| E6XLE5 | – | – | – | – | – | – | – | + (88.25%) |
| Q8GJK1 | – | – | – | – | – | – | – | + (99.30%) |
| Q6ZYR2 | – | – | – | – | – | – | – | + (99.71%) |
| A4Y1U2 | – | – | – | – | – | – | – | + (78.41%) |
| A4Y1U5 | – | – | – | – | – | – | – | + |
| A1KQX7 | – | – | – | – | – | – | – | + |
| A4Y1U4 | – | – | – | – | – | – | – | + |
| A4YB89 | – | – | – | – | – | – | – | + |

Note:
Fragment presence and correct amplification size is denoted with + and absence is denoted with –. When the fragment was sequenced, the percent of identity with the Pdp11 genome is included.

proteins due to the presence of sequence gaps within coding regions. As a consequence, a negative result, that is, the lack of a protein, may not warrant its absence. That is why the genome comparisons is mainly focused on present proteins. Consequently, genome sequences of 15 out of the 16 Pdp11-specific, complete proteins of Table 5 where present in Pdp11 but absent in NPSs (Table 6), only trimethylamine *N*-oxide reductase being present in one of the NPS groups (SH4-SH12). This experimental confirmation of TarSynFlow predictions using genome drafts confirms robustness and reliability of the algorithm.

The algorithm underlying TarSynFlow provides not only a presence/absence pattern, but also functional information about the genome specific proteins that can be exploited to focus further experimental research in regions that are related with the biological problem of interest. Therefore, TarSynFlow may be useful in synteny studies more focused on functional conservation (as in Fig. 2) than in sequence conservation. Moreover, TarSynFlow is so flexible that can focus the analysis only on a small sequence pool of interest, or extend the comparison of several genomes against the same reference. Since gene presence and position, and functional annotation were provided, interpretation of the biological problem is facilitated.

## Apparent clustering of non-probiotic strains

Non-probiotic strain genome drafts in Table 3 have a genome size and %GC absolutely compatible with other *Shewanella sp.*. Number of scaffolds, N50 and the completeness
estimation was suitable for genome comparisons using TarSynFlow. Phenotypic characterization of NPSs (Table 1) produced separate combinations of LD50, RAPD bands and growing patterns (*Esteve, Merchán & Alcaide, 2016*), demonstrating that the seven independent isolates can be considered different strains. However, specific and shared proteins (Table 4) among them suggested that pathogenic strains could be tentatively classified into two groups, one consisting in SH6, SH16 and SH9, and the other containing SH4 and SH12. Nevertheless, saprophytic strains do not conform a group, since SdM1 is closer to the SH6-SH9-SH16 group and SdM2 is closer to Pdp11 (Fig. 3). Even though this classification is not supported by the RAPD profiles, we suggest that presence/absence of genes regarding the probiotic, pathogen and saprophytic character of *Shewanella putrefaciens* are more significant than non-specific patterns of repetitive sequences.

## Pdp11-specific proteins may explain some probiotic characteristics of Pdp11

The comparison of the finished genome of Pdp11 with seven NPS genome drafts can find present and absent proteins, but cannot distinguish rearrangements from misassembling. Genome rearrangements cannot be used then to explain the differences between probiotic and non-probiotic character in this work. With this in mind, a total of 64 proteins are absent in Pdp11 but shared by all pathogenic and saprophytic strains (File S2). Since the Pdp11 genome is finished, such an absence suggests that these 64 proteins are not involved at all in the probiotic character of Pdp11, even though the functional annotation for these proteins gives no clear clue to explain their absence in a probiotic strain.

More promising information is expected from the 19 Pdp11-specific proteins of Table 5. To avoid confounding results, only the 16 complete proteins of Table 5 were validated by PCR amplification in the seven NPSs of Table 1. All but O86914 received the experimental confirmation of their presence in Pdp11 and their absence in the NPSs (Table 6; Fig. 4). This suggests that these 15 Pdp11-proteins might provide some probiotic benefits and their detailed functional inspection may shed light on the molecular basis of some probiotic features of Pdp11.

## Pdp11 seems to have undergone some genomic rearrangements

The acquisition or loss of genetic material by the horizontal exchange of mobile genetic elements such as plasmids, phages, transposons and integrative and conjugative elements (ICEs) has been demonstrated to be essential to allow microorganisms the adaptation to new niches (*Aminov, 2011*; *De Maayer et al., 2015*). In addition, some *Shewanella putrefaciens* strains revealed a mosaic element of plasmid, phage and transposon-like sequences typical of ICEs, which was related to resistance to heavy metals (*Pembroke & Piterina, 2006*). ICEs can encode factors involved in the resistance to antimicrobials and in the production of secondary metabolites such as antimicrobials (*Burrus, Marrero & Waldor, 2006*), whereas transposons have made major contributions allowing to bacteria acquiring additional genetic information, including numerous

metabolic genes (*Nicolas et al., 2015*), or playing important regulatory roles (*Szuplewska et al., 2014*). This fact could improve the versatility for the probiotic strain to compete with the gut microbiota for metabolic resources and increase its opportunity to colonize the intestine. Interestingly, half of Pdp11-specific proteins in Table 5 can be directly related to these processes, such as integrase (Q6ZYR2), resolvase (A4Y1U2), transposases (A4Y1U5, A4YB89), excisionase (A1KQX7), a regulator for conjugative element (Q8GJK1), a plasmid stabilization system (A4Y1U4) and proteins for conjugative transfer of DNA (A4L329, Q70IK8, Q70IK5). The presence of such Pdp11-specific genes can explain why Fig. 2 shows some sequence rearrangements in Pdp11 with respect to SdM2 and opens the door to a future study related to the repercussion of mobile elements in Pdp11 genome.

## Pdp11 might have gained genes promoting colonization

Gastrointestinal tract is a stressful environment where the probiotic cells, to survive, have to respond to and thrive under a variety of extreme conditions, such as crossing the stomach, presence of bile salts, a vast array of microorganism inhabits, antimicrobials (of both host and bacterial origin), etc. (*Brunke & Hube, 2014*). For this reason, the colonization capability is an important probiotic feature since they should be able to show a good tolerance to intestinal stress (*Parente et al., 2010*). The high colonization capability of Pdp11 in *Solea senegelansis* gut has been already demonstrated in our group (*Tapia-Paniagua et al., 2014*, *2015*). Colonization capability is not expected to depend on single features but on a set of abilities such as adhesion, obtaining of nutrients, survival in presence of bile salts, and competition with the other microorganisms.

The presence of Q8GJK1, a HTH-type transcriptional regulator for conjugative element pMERPH, may be explained by the fact that it contains one of the most common motifs observed in DNA-binding proteins controlling a wide range of functions such as DNA repair and replication, RNA metabolism and protein-protein interactions in diverse signaling contexts. Several of these bacterial regulators are repressors of genes and operons for membrane transport and cell envelope permeability involved in the resistance to antibiotics, bacteriocins and host-encoded antimicrobials (*Grkovic, Brown & Skurray, 2002*). A very high number of different microorganisms are populating guts, and most of them have the capability to produce antibiotics or bacteriocins (*Dicks et al., 2018*). Hence, the Pdp11-specific ICEs and HTH regulators in Table 5, previously described as sources of drug resistance genes (*Peters et al., 1991*; *Fang et al., 2018*), together with the demonstration that Pdp11 (*Chabrillón et al., 2005*) and other *Shewanella sp.* (*De la Rosa-Garca et al., 2007*) can produce antimicrobial substances, including bacteriocins (*Cimmino, Olaitan & Rolain, 2016*), might support the improved ability of Pdp11 to colonize the intestinal environment.

One of the digestive stresses are bile salts, which are detergent-like molecules with bactericidal effect. Bacteria usually thwart their lethal effect by limiting the entry of bile salts into the cell by active efflux transports (*Alvarez-Ortega, Olivares & Martnez, 2013*). Pdp11 resistance to bile salts was already reported (*Chabrillón et al., 2006*), and the efflux pumps of the resistance-nodulation-division family present in many Gram-negative

bacteria (*Opperman & Nguyen, 2015*) was invoked as the source of such resistance. Moreover, this type of efflux pumps have an important role in the capability of colonization of certain microorganisms by resistance to bile salts (*Alvarez-Ortega, Olivares & Martnez, 2013*; *Anes et al., 2015*). Interestingly, among the Pdp11-specific proteins of Table 5 there are two cases of bile acid/detergent exporters, one of them encoding for the membrane fusion component VexC (E6X1Z3) and the other for the permease component VexD (E6X1Z2). Their presence can support the previous findings (*Chabrillón et al., 2006*), and is consistent with earlier reports that demonstrate that VexCD efflux system had an important role in the resistance to *Vibrio cholera* to bile salts (*Bina et al., 2006*).

There is another cell envelope component involved in the bile resistance: the enterobacterial common antigen (ECA) located in the outer leaflet of the outer membrane (*Ramos-Morales et al., 2003*; *Urdaneta & Casadesús, 2017*). Table 5 contain two members of this transport family (E6XG15 and E6XG14) that have undecaprenyl-phosphate α-*N*-acetylglucosaminyl 1-phosphate transferase activity that catalyzes the transfer of the GlcNAc-1-phosphate moiety from UDP-GlcNAc onto the carrier lipid undecaprenyl phosphate. This is the first lipid-linked intermediate involved in ECA synthesis, and an acceptor for the addition of subsequent sugars to complete the biosynthesis of *O*-antigen lipopolysaccharide. Very likely, these transferases could improve the repair of damages caused by the bile salts and increase the resistance against bile salts showed by this probiotic microorganism. In conclusion, VexC (E6X1Z3), VexD (E6X1Z2), E6XG15 and E6XG14 can explain together the bile salt resistance phenotype of Pdp11 and are tempting candidates to promote at least some part of the gut colonization capability of this strain.

Another Pdp11-specific protein of Table 5 is E6X1Z5, an Mlta-interacting MipA family protein, that could allow the response to changes in the intestinal conditions by facilitating the assembling of the complex implied in the synthesis of murein sacculus that stabilizes the cell envelope of Gram-negative bacteria. The metabolism of murein involves the specific interaction of several proteins and the formation of a multienzyme complex of murein synthases and hydrolases which shows a highly degree of variability (*Von Rechenberg et al., 1996*; *Romeis & Höltje, 1994*) to allow cell survival even in the case of spontaneous mutations in some of these proteins (*Vollmer, Von Rechenberg & Höltje, 1999*). The multienzyme complex contains proteins of MltA-interacting MipA family, where MipA is considered a structural protein mediating the assembly of MltA to PBP1B into a complex (*Vollmer, Von Rechenberg & Höltje, 1999*). Additionally, MipA was identified as a protein related to antibiotic resistance in strains of *Escherichia coli* (*Zhang et al., 2015*). In addition, MipA protein has been demonstrated to interfere the adherence of enterotoxigenic *E. coli* strains (*Hays et al., 2016*) and it could be related to the capability showed by Pdp11 to inhibit the adhesion to intestinal mucus of *Solea senegalensis* of pathogen such as *Vibrio harveyi* and *Photobacterium damselae* subp *piscicida* (*Chabrillón et al., 2006*). Taking together, the E6X1Z5 member of MipA not only can help in the capability of Pdp11 to grow in presence of high levels of bile salts, but also can play a role in the improved protection against the antimicrobial compounds

present in gut (*Gillor, Etzion & Riley, 2008*). Therefore, this protein can have a supporting role in the colonization capability of Pdp11.

Three proteins of Table 5 could not be identified, but one of them, E6XL69, is known to be an integral membrane protein. This prompted us to think that it could be another membrane transporter that might contribute to resistance to bile salts, antimicrobial or any other kind of gut stress. However, more work is required to identify the roles of these three unknown Pdp11-specific proteins.

Finally, even if O86914, a trimethylamine-*N*-oxide (TMAO) reductase, appears in Pdp11 as well as in the group of pathogenic strains SH4-SH12 (Table 6), it merits some attention since it can contribute to the probiotic character of Pdp11. This enzyme can allow growth under anaerobic conditions using trimethylamine oxide (TMAO, a major low molecular mass constituent of marine fish *Barrett & Kwan, 1985*) as an alternative terminal electron acceptor. Therefore, it could also be related with the capability of gut colonization showed by Pdp11.

## CONCLUSIONS

Genome drafts with ≥99.1% gene completeness from seven new NPSs of by *Shewanella putrefaciens* were obtained (Table 3). Genome sizes and GC contents of NPSs are in agreement with those of other finished *Shewanella sp.* in NCBI's genome database, indicating the apparent genome homogeneity of this group. Nevertheless, NPSs seem to cluster in two main groups, one containing SH4 and SH12, and the other with SH6, SH9 and SH16 (Fig. 3). Draft genome closure would be desirable in a near future but here it is demonstrated that their current completeness can guarantee that an absent gene in a NPS really implies its absence. In fact, closed genomes might also help to explain the clustering emerging from Fig. 3 and Table 4.

Pdp11 and NPS genomes were suitable for testing TarSynFlow algorithm (Fig. 1) in the seek of common and differential genes. As a proof of concept, a curated set of 3,774 UniProtKB proteins from *Shewanella sp.* (File S1) allowed the comparison of one probiotic finished genome (Pdp11) with newly assembled genome drafts. The dense and reliable set of links between genomes (Fig. 2) supports the hypothesis that Pdp11 underwent some specific genome rearrangements spurred by ICEs and transposons, as well as plasmid exchanges with other bacteria. These rearrangements allowed the recovery of Pdp11-specific proteins and NPS-specific proteins (Table 4), and studies on their tentative contribution to the probiosis of Pdp11. Potential correlation between the rearranged regions and the Pdp11-specific genes or transposon jumping would facilitate any future experimental exploration of putative probiotic genes.

A total of 15 genes were found exclusive of Pdp11 (Tables 5 and 6), that is, strong candidates to be probiotic-specific proteins. Their presence in Pdp11 as well as their absence in NPSs was experimentally illustrated (Fig. 4) to dispel any doubt derived from the drafting nature of NPS genomes. Functional inspection of the 15 probiotic-specific proteins reveals that most of them could improve gut colonization capabilities and inhibit pathogen adhesion to the intestinal mucus of *Solea senegalensis*. For example, Pdp11 can grow on high bile salt content based on exclusive VexD, VexC, MltA-interacting

MipA and transferases, while ICEs and HTH regulators can help to gain antimicrobial substance production as well as antibiotic and bacteriocin resistances. The mere transformation of one to all gene candidates in a NPS will not assure that the resulting transformant would be probiotic, since nothing is said here about the genes that are lacking in Pdp11 that prevent any possible pathogenicity. Therefore, searching for Pdp11 absent genes would merit the effort since they would account for its non-pathogenic nature, even though demonstrating an absence is more difficult that demonstrating a presence.

As a result, even if some probiotic features of Pdp11 have been revealed, the presence of uncharacterized proteins in Table 5 and File S2 indicates that the task of defining the molecular bases of probiosis is far from being resolved. All together, the results concerning Pdp11-specific proteins both support the algorithmic design of TarSynFlow and illustrate its suitability for comparisons between several bacterial genomes, whether the genomes are finished or fragmented.

## ABBREVIATIONS

| | |
|---|---|
| **cfu** | colony forming units |
| **ICE** | integrative and conjugative element |
| **LD50** | lethal dose 50 |
| **ND** | not determined |
| **NPS** | non-probiotic strain |
| **PM** | putative match |
| **RM** | reliable match. |

## ACKNOWLEDGEMENTS

The authors also thankfully acknowledge both the Ultrasequencing Unit and the computer resources and technical support provided by the Plataforma Andaluza de Bioinformática of the University of Málaga.

### Funding
This work has been co-supported by the ERDF (European Regional Development Fund) 2014–2020 and the Spanish Research Agency (AGL2017-83370-C3-3-R) and INIA (RTA2017-00054-C03-03) grants. There was no additional external funding received for this study. The funders had no role in study design, data collection and analysis, decision to publish, or preparation of the manuscript.

### Grant Disclosures
The following grant information was disclosed by the authors:
ERDF: European Regional Development Fund 2014–2020.
Spanish Research Agency: AGL2017-83370-C3-3-R.
INIA: RTA2017-00054-C03-03.

## Competing Interests

The authors declare that they have no competing interests.

## Author Contributions

- Pedro Seoane performed the experiments, analyzed the data, contributed reagents/materials/analysis tools, prepared figures and/or tables, authored or reviewed drafts of the paper, approved the final draft.
- Silvana T. Tapia-Paniagua performed the experiments, analyzed the data, contributed reagents/materials/analysis tools, prepared figures and/or tables, authored or reviewed drafts of the paper, approved the final draft.
- Rocío Bautista performed the experiments, analyzed the data, contributed reagents/materials/analysis tools, authored or reviewed drafts of the paper, approved the final draft.
- Elena Alcaide performed the experiments, contributed reagents/materials/analysis tools, approved the final draft.
- Consuelo Esteve performed the experiments, contributed reagents/materials/analysis tools, approved the final draft.
- Eduardo Martínez-Manzanares conceived and designed the experiments, performed the experiments, contributed reagents/materials/analysis tools, approved the final draft.
- M. Carmen Balebona conceived and designed the experiments, contributed reagents/materials/analysis tools, authored or reviewed drafts of the paper, approved the final draft.
- M. Gonzalo Claros conceived and designed the experiments, analyzed the data, contributed reagents/materials/analysis tools, prepared figures and/or tables, authored or reviewed drafts of the paper, approved the final draft.
- Miguel A. Moriñigo conceived and designed the experiments, analyzed the data, contributed reagents/materials/analysis tools, prepared figures and/or tables, authored or reviewed drafts of the paper, approved the final draft.

## Data Availability

Code: https://github.com/seoanezonjic/TarSynFlow.

Shewanella sp. Pdp11 chromosome, complete genome: AC# CP015194.1.

The rest of *S. putrefaciens* raw reads: PRJNA510237.

## Supplemental Information

Supplemental information for this article can be found online at http://dx.doi.org/10.7717/peerj.6526#supplemental-information.

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
