# Peer review of "TarSynFlow, a workflow for bacterial genome comparisons that revealed genes putatively involved in the probiotic character of Shewanella putrefaciens strain Pdp11"

_PeerJ, doi:10.7717/peerj.6526_

## Round 0.1 · original submission · Minor Revisions

Dear Dr. Seoane and colleagues:

Thanks for submitting your manuscript to PeerJ. I have now received two independent reviews of your work, and as you will see, both are very favorable. Well done! Nonetheless, both reviewers raised some relatively minor concerns about the research, and areas where the manuscript can be improved. I agree with the reviewers, and thus feel that their concerns should be adequately addressed before moving forward.

Therefore, I am recommending that you revise your manuscript accordingly, taking into account all of the issues raised by the reviewers. I do believe that your manuscript will be ready for publication once these issues are addressed.

Good luck with your revision,

-joe

Reviewer 1 ·

Basic reporting

'no comment'

Experimental design

'no comment'

Validity of the findings

'no comment'

Additional comments

The manuscript entitled “TarSynFlow, a workflow for bacterial genome comparisons that revealed genes putatively involved in the probiotic character of Shewanella putrefaciens strain Pdp11”, by Seoane et al., presents an interesting bioinformatic tool (Targeted Synteny Workflow) for comparative genomic analysis. The authors show that the application of such tool for genome comparison between probiotic vs non-probiotic strains is of upmost importance, and might allow to identify probiosis related genes, further detailing current knowledge on probiotics mechanisms of action.
The manuscript is very well written, the methodologies used are adequate and well described, with exception of a few clarifications that authors should do, to improve the overall quality of the manuscript before acceptance for publication.

1. The authors can add some justification to sequence just six out of the 16 gene amplicons analyzed, and also tentatively comment on the low identity percentage obtained for 2 of the sequences, namely 88.25 and 78.41%.

2. In the bottom of Figure 4, the protein code, rather than the protein function, should be used to maintain the same nomenclature throughout the paper and facilitate the comparison with Tables 2, 5 and 6.

3. Line 106: please correct the “dose at which 50% of bacteria die”, since are not the bacteria who die

4. Line 118: please check the code provided PRJNA510237, since no its can be found at the BioProject database at NCBI

5. Line 221: please replace the word “goodness” for a more scientific one such as potential, applicability, effectiveness, etc

6. Line 326: please correct “Nevetheless” to Nevertheless

7. Line 352: “are” at the end should be deleted

8. Line 371: “competence” should be replaced by “competition”

9. Lines 380-382: the authors should add some reference(s) to support the sentence describing Pdp11 strain as being resistant to bacteriocins

·

Basic reporting

The paper is generally well written. The statements and arguments are presented in a clear manner. The figures are clear and easy to follow.

Experimental design

The research question and hypothesis are clearly presented. Deatils are available for this paper to be reproduced. This is particularly important since the manuscript is describing a a method or approach.

Validity of the findings

The method presented here is highly valuable for the progress of probiotics research in fish. The data are presented in a highly understandable manner. The implications are logical and are supported by the data.

Additional comments

Figure 1. a more descriptive caption would improve this figure. At present, it is too short and vague. The reader needs to scan through the text in order to understand the figure.

Table 4 is best presented through a venn digram.

---

## Round 0.2 · accepted · Accept

Dear Dr. Seoane and colleagues:

Thanks for revising your manuscript based on the concerns raised by the two reviewers. I now believe that your manuscript is suitable for publication. Congratulations! I look forward to seeing this work in print, and I anticipate it being an important resource for communities studying methods in comparative genomics and gene targeting. Thanks again for choosing PeerJ to publish such important work.

Best,

-joe

#